# Demand Creation for COVID-19 Vaccination: Overcoming Vaccine Hesitancy through Social Marketing

**DOI:** 10.3390/vaccines9040319

**Published:** 2021-04-01

**Authors:** William Douglas Evans, Jeff French

**Affiliations:** 1Milken Institute School of Public Health, The George Washington University, Washington, DC 20052, USA; 2Department, Strategic Social Marketing Ltd., Atabara, Condors, Liphook, Hampshire GU30 7QW, UK; jeff.french@strategic-social-marketing.org

**Keywords:** COVID-19, vaccination, vaccine hesitancy, demand creation, health communication, social marketing, branding, behavioral economics

## Abstract

The COVID-19 pandemic has led to millions of deaths and tested the capabilities of the medical and public health systems worldwide. Over the next two years as more approved vaccines are made available and supply meets or exceeds demand, medical and public health professionals will increasingly be faced with the challenge of vaccine hesitancy. There is an urgent need to create demand in groups that are either uninformed, vaccine hesitant, or actively resistant to COVID-19 vaccination. This study reviews theory, evidence, and practice recommendations to develop a vaccine demand creation strategy that has wide applicability. Specifically, we focus on key elements including supply side confidence, vaccine brand promotion strategy, service marketing as it relates to vaccine distribution, and competition strategy. We present evidence that these strategies can make a significant contribution to overcoming COVID-19 hesitancy in a high supply scenario. The paper also makes recommendations about factors that need to be considered in relation to vaccine delivery services and systems that, if done badly, may reduce uptake or result in the creation of more vaccine hesitancy. In summary, there is a need for well researched and tested demand creation strategies that integrate with brand strategy, supply side, and service delivery.

## 1. Introduction

The COVID-19 pandemic has led to millions of deaths and tested the capabilities of the medical and public health systems worldwide. As more approved vaccines are made available and supply meets or exceeds demand, medical and public health professionals will increasingly be faced with the challenge of vaccine hesitancy [1,2]. There is an urgent need to create demand in groups that are either uninformed, vaccine hesitant, or actively resistant to COVID-19 vaccination. This paper reviews theory, evidence, and practice recommendations from the fields of health communication, health promotion, health education, behavioral economics, and social marketing to develop a vaccine demand creation strategy.

Specifically, we focus on key elements including supply side confidence [3], vaccine brand promotion strategy [4,5], service marketing [6], and competition (i.e., the promotion of unproven alternatives to vaccination) [7]. We recommend a demand creation strategy for COVID-19 vaccination based on a combined approach of building a positive brand identity for vaccination behavior together with the use of proven behavioral sciences approaches (e.g., nudges) [8,9] combined with strategic social marketing planning. Success will also be dependent on the brand characteristics and brand equity of the vaccine products themselves, which differ and have received varying levels of coverage both positive and negative in traditional news and social media [10].

We present a prototype COVID-19 brand identity based on the reality that there are multiple vaccine brands in the market and a need for specific marketing to build confidence and demand for them both collectively and for individual vaccines. Such communications and marketing approaches aimed at persuasion should follow World Health Organization guidance, outlined in the paper [11]. Within the suggested strategy, we also include a review of and recommendations for the use of both incentives and disincentives that can augment other efforts and give examples of interventions that can misfire when using incentives and disincentives [12]. The paper also makes recommendations about factors that need to be considered in relation to vaccine delivery services and systems that if done badly may reduce vaccine uptake or result in the creation of more vaccine hesitancy. We argue for tactics, including formative research (i.e., to create more effective vaccine promotion strategies and tactics), to avoid common failings of incentive schemes that can arise if the degree of and reasons for specific population vaccine hesitancy is not understood in advance [13]. In summary, we argue for the need for well researched and tested demand creation strategies that use a mix of interventions that integrate communication campaigns with brand strategy, supply side issues, competition strategy, and service delivery considerations.

### What We Know Works

Previously, the authors published a review of key guidelines drawn from the field of social marketing and related behavior change frameworks for COVID-19 vaccine promotion [14]. French et al. (2020) identified 10 actions that governments, public health agencies, and other community actors can take to increase acceptance and demand for COVID-19 vaccination, based on lessons learned from the field of social marketing and behavior change interventions. These key actions are based on evidence from the field broadly and vaccination promotion in particular. Specifically, the 10 key actions recommended are: (1) public trust building and community engagement, (2) vaccine demand building, (3) audience targeting and segmentation, (4) asset mapping, mobilization and coalition building, (5) planning and implementation, (6) competition and barrier analysis and action, (7) messaging and promotions strategy, (8) traditional media management, (9) digital media management, (10) vaccine access (supply) [14]. The paper pointed out that governments and relevant bodies can implement these key evidence-based processes and in so doing enhance vaccine uptake. The paper also noted the importance of local context and population-specific barriers and opportunities in implementing a vaccine demand creation strategy.

Being clear about those key lessons from social marketing for vaccine promotion is an important first step, and helps to inform a broad strategic approach to promoting vaccine uptake. This paper takes the next step by identifying, and providing evidence for, specific strategies to reduce vaccine hesitancy. Specifically, we propose evidence-based actions that governments, public health agencies, and community-based organizations can take to create demand based on an analysis of the degree and basis of vaccine hesitancy in specific population groups (segmentation). We propose a strategy for “what works” to design and execute an appealing brand identity for COVID-19 vaccines.

Brands build relationships between consumers and products, services, or lifestyles by providing beneficial exchanges and adding value to their objects [15]. Effective branding clearly frames the choices facings consumers and creates an identity that promotes brand choice over alternatives. In the health domain, this encompasses both product (e.g., vaccine) choice as well as behavior change (e.g., choosing to get vaccinated) [16]. Vaccine branding is needed for short term promotional purposes and to inform long-term strategic planning in both high supply and low supply situations.

The early stages of vaccine rollout in many countries have been characterized by an emphasis on building confidence in vaccines generically and appeals to social solidarity (i.e., getting vaccinated is normative, or the *right* thing to do, and that vaccines are safe and effective). In addition, governments have also focused on making access easy, and at low or no cost. However, as time goes on and supply is in equilibrium or exceeds demand considerations beyond safety, effectiveness and social solidarity will be needed to promote uptake among the hesitant [17].

Diffusion of innovation (DOI) theory offers guidance here [18]. We envision a diffusion curve ranging from the early adopters who actively seek vaccination, to those who are unsure or hesitant, to those who actively resist and or are actively opposed to vaccination. In the short-term, COVID-19 demand creation will be relatively easy as the early adopters, (i.e., the enthusiastic) will come forward without much persuasive effort. However, as time goes on, in high-supply countries, those who are hesitant will come to represent more of the unvaccinated. As the hesitant come to the front of the queue there will be a need for more targeted approaches that speaks to the concerns held by these groups.

The same is true for low-supply countries, with the difference being the hesitant as a proportion of the community. One way to delay lower demand in both situations is, in the short term, to adopt a selective targeting strategy aimed at specific high-risk groups who also exhibit high demand (e.g., older adults with co-morbidities and/or front-line health workers).

Brand development always occurs in the context of competition and barrier analysis (another of the ten key guidelines set out above). Choosing one product, or service, or health behavior over another involves comparisons of costs and benefits, and overcoming barriers to purchase, adoption, and the value associated with the product, service or behavior [19]. For COVID-19 vaccination, this will critically involve where a population stands on the diffusion curve for vaccine uptake from enthusiastic to resistant.

Inevitably, as the vaccine rolls out and the remaining unvaccinated population is increasingly hesitant and resistant, vaccine demand creation will need to move from the generic “get vaccinated” branding towards a more specific vaccine product brand promotion approach. In a competitive and potentially crowded market place with many vaccines behind developed and approved we will enter a situation in which people in some countries (mainly the richer ones) will potentially have an individual choice of vaccine products each with varying reported levels of effectiveness, side effects, and other potential factors that may influence both preference and hesitancy. Before we reach this stage, governments and public health agencies must make decisions about which vaccines to procure for their population. Decisions about which vaccines to procure may well need to be tied increasingly into target group confidence in particular vaccine brands as well as maintaining confidence in a generic concept of the benefits of vaccination.

In brief, we need a clearly identified generic demand strategy and then increasingly specific brand promotion strategies to overcome hesitancy. Based on these market-level considerations, we propose a conceptual model for approaching and overcoming COVID-19 vaccination hesitancy (see Figure 1).

This model is based on prior vaccination research efforts that recognizes the mediation process leading to uptake [20]. In this model, vaccine uptake is mediated by changes in vaccine beliefs and intentions delivered through COVID-19 promotional campaign exposures (e.g., consumption of advertisements and other content on media). This process of demand creation through promotional efforts occurs in the context of social support from government and civil society, service and product availability, health care system structures, and economic factors that may promote or inhibit vaccine uptake. Success in overcoming hesitancy must recognize that demand creation occurs within these marketplace system dynamics and account for them in branded vaccination campaigns.

One critical factor in demand creation is the point of decision for vaccination in the health care setting. Health care workers (HCW) including doctors, nurses, and community-based HCW must demonstrate that they have been vaccinated and support vaccine uptake in the population in order to build public trust. Promoting vaccine uptake and building trust *among HCW* in the COVID-19 vaccine is thus a crucial prerequisite to a broad, population-based demand creation strategy.

Another import factor in the model is support from elected officials and other public figures with major media platforms to support vaccination. In the US, figures such as President Joseph R. Biden and Dr. Anthony Fauci have championed vaccination in speeches and made public appearances receiving one of the vaccines on television. In New Zealand, which has had remarkably low incidence of COVID-19 infection, Prime Minister Jacinda Ardern has received strong support for her support for public health measures and vaccination.

On the other hand, public figures can also play a negative role inflaming vaccine resistance through public discourse (https://www.thesun.co.uk/news/14131729/germany-france-covid-thousands-refuse-oxford-jab-scaremongering/) (accessed on 22 March 2021). For example, French President Emmanuel Macron in late January 2021 made negative comments about the effectiveness of the Oxford University Astra Zeneca vaccine (Astra Zeneca, Oxford, UK) among older people, which were subsequently proved false. Confidence across Europe was further reduced in March 2021 when a number of countries, including France and Germany, acting against the advice of the European Medicines Agency (EMA) and WHO, decided to suspend the use of the vaccine due to a small number of flagged health events related to blood-clotting. Again, after reviewing the evidence, the EMA declared that there was no evidence of a causal relationship between the vaccine and the observed blood-clotting incidents was not established. This lack of consistency and over-zealous precautions resulted in a lack of confidence in France and many other European countries in the Oxford vaccine, despite trials indicating that it is one of the safest and most effective vaccines in existence, as well as the cheapest and most easily transported. However, once a negative impression has been given it is very difficult to remove it from the public consciousness.

Social media plays a critical role here, as there are significant vaccine hesitant and resistant communities on platforms such as Facebook, and celebrities and online social influencers who play a role in promoting resistance [21]. For example, the Centre of Countering Digital Hate (Center for Countering Digital Hate (counterhate.com) (accessed on 22 March 2021) not-for-profit organization track over 400 anti-vaccine accounts across the big social media platforms. The center has found that such accounts have over 60 million followers with many more people joining every day. Many of these sites represent entrepreneurs with product to sell such as ‘homeopathic immunization’. About half of the accounts are conspiracists who profit from online advertising revenue. Such social media groups deploy three key tactics, (1) sowing doubt about the existence and seriousness of the pandemic (2) spreading concern about safety, (3) stressing the untrustworthiness of experts. A number of social media providers such as Facebook are now seeking to reduce the impact of negative social media coverage of anti-vaccination groups but to date, social media continues to be a primary source of dis- and misinformation in relation to vaccination of all types.

We must envision a process of brand development to promote generic short-term COVID-19 vaccination and overcome hesitancy while the vaccine rollout is in its early stages, and also a long-term, vaccine product specific strategy as we reach the hesitant and resistant populations. The question becomes, “How do we develop and test the proposed model and increase evidence on how to reduce hesitancy long-term?” The remainder of this paper addresses that question and unpacks specific strategies for COVID-19 vaccination branding within a context of evolving market dynamics.

## 2. Materials and Methods

### 2.1. Vaccine Hesitancy Intersection with Supply

In the early phases of vaccination in 2021–2022, there will in many countries be more demand than supply. However, as time passes and there is more supply than demand, hesitancy will increasingly be a key (if not the most important) factor in achieving global population protection. Hesitancy has many causes and the hesitant are not a uniform group, reasons vary by context and history. As many vaccines emerge and are approved the potential to add to hesitancy will emerge in the form of preferences for one vaccine over another and depending on in-country availability hesitancy may be increased if access is not available to what citizens believe are the post effective/safe vaccines due to issues such as cost or political alignment between suppler counties and receiving countries.

### 2.2. Inequality/Disparity in Vaccine Supply and Uptake within and between Countries

COVID-19 has the potential to reverse many of the advances seen in recent decades in reducing inequality and promoting wellbeing both within and between rich and low and middle in countries. One of the key disparities is between rich and poor countries speed of access to vaccines [22]. Both cost and supply will factor in in-country availability but so will confidence in which vaccine is acquired for mass distribution [23]. Such vaccines will need to be safe, effective, practice (e.g., little need for a cold chain) and have a strong trusted brand. For example, it is unlikely that in countries where Russia is perceived to be a less trustworthy country that people will be happy to uptake the Sputnik V vaccine, regardless of its efficacy. Similarly, in countries such as China that want to promote their own vaccine developers, it is unlikely that people will have high demand for Western vaccine brands such as Pfizer or AstraZeneca.

### 2.3. Available Vaccines

The speed of COVID-19 vaccine development in 2020–2021 was unprecedented [24]. As of the time of this writing, there are a growing number of vaccines approved and in many more in the development pipeline. The sheer number of vaccines that will be available will lead to a need to discriminate which vaccine is most appropriate in a given country setting. Vaccine brand characteristics such as effectiveness, type of regulatory approval, need for cold storage, price, viability, security of supply, origin of manufacture, time of supply, public confidence, and professional confidence will all play into decisions about which vaccines governments will seek to acquire and distribute. Factors such as public trust and affiliation with specific brands will also need to be considered.

### 2.4. Developing a Market-Based Model for COVID-19 Vaccine Promotion

Exchange theory recognizes that brands create identities that offer specific benefits to consumers within a range of choices, whether they be products, services, or behaviors [25,26]. This can be viewed as a form of choice architecture, or the design of different ways in which choices can be presented to consumers, and the impact of that presentation on consumer decision-making and behavior [27,28]. Optimizing choice architectures for generic vaccination decision making early in the roll out and increasingly brand-specific decision making over time is crucial to vaccine uptake.

The Porter market forces model provides a basis for conceptualizing how a vaccine can enter into the market and how it can be positioned so that it is accepted and demanded by consumers [25]. Such positioning leads to competitive advantage for the vaccine. There are two basic types of competitive advantage an organization can possess: cost or differentiation [25]. The two basic types of competitive advantage combined with the scope of activities for which a firm seeks to achieve them, lead to three generic strategies for achieving above average performance in an industry: or in the case of vaccines uptake rates. The three strategies are: (1) cost leadership; (2) differentiation; (3) focus. The focus strategy has two variants: cost focus, and differentiation focus.

Figure 2 illustrates a simple model of vaccine choice architecture that could be used to inform the development of a vaccination promotion brand strategy.

The figure may be interpreted as defining the interaction of choice as a demand driver with product supply. In the maximum choice scenario, there is high access to vaccine and multiple products, so the greatest number of consumer options and decisions to be made. In this scenario, brand promotion strategy and tactics must focus on personal benefits and characteristics of specific vaccines and social solidarity. In this situation, public health authorities may wish to emphasize the individual strengths of specific vaccines if citizens are afforded a choice of which vaccine to have and or emphasize just how easy, fast, and convenient access to vaccines is in the country or region.

For the easy access/low choice scenario, there is at least one vaccine option is readily available, but few or no product choices. Here, the objective is to promote the advantages of a selected vaccine for the country and community and focus on building trust in COVID-19 vaccination overall and specifically the available product. In this situation, governments and public health organization would do well to emphasize the strengths of the available vaccine in comparison with other brands and ease of access due to plentiful supply.

For the low access but multiple-choice scenario, there will be barriers to product availability due to distribution and/or supply limitations. Here, strategy and tactics must focus on priority groups (e.g., frontline workers and older adults with co-morbidities) and raise trust in all available choices while explaining the long-term plan to increase access.

Finally, in the low access and minimum choice scenario, all options are limited and the focus of strategy and tactics must be long-term. The aim should be to promote uptake of available vaccines among high-risk groups and explain the longer-term societal strategy to promote wider access. In this situation, there will also be a need to manage citizen expectation about future vaccine access by setting out timetables for access that can be delivered.

### 2.5. Competition Strategy

There are four main types of competition with vaccine promotion that must be addressed in developing a generic vaccine promotion brand at any stage of product roll out. First, there is the passive competition of simple indifference, lack of concern, and inertia as barriers to action. The population is essentially in a pre-contemplation stage of thought with respect to the vaccine; it is not motivated to learn more about it or to enter into a choice architecture scenario [29]. Passive competition can also be viewed as a manifestation of lack of concern. Interventions aimed at raising concern without developing undue fear should be considered as we know that people will only act to protect their health when they perceive threats to be real, significant and likely. Reminding citizens of the risks associated with not being vaccinated is a legitimate form of intervention in this situation.

Second, there is competition in the form of “mis-information,” such as mistaken comments about vaccine efficacy, effectiveness, side-effects, other protective actions such as social distancing or mask wearing, or personal or societal risks from COVID-19. These misstatements are especially problematic when they come from highly visible public officials or other figures such as celebrities, and must be countered by fact-based correct information as illustrated above.

Third, there is disinformation, such as anti-vaccine advocacy and active promotion of known false narratives in order to create a negative public opinion and build hesitancy and resistance to vaccination. For example, the statements by US President Donald Trump to the effect that consuming bleach would act as a disinfectant for COVID-19 became a sensation online and on social media platforms and spread dis-information (https://www.bloomberg.com/news/articles/2020-04-25/-disinfectant-bleach-tweets-top-covid-19-after-trump-gaffe) (accessed on 22 March 2021).

Fourth, there is the complex competition of vaccine rivalry and nationalism drivers between competing vaccine producers located in different countries such as the USA, UK, China, Russia, etc. In some instances, these nationalistic tendances can be positive forces to promote vaccination *within* the country producing a given vaccine brand (e.g., pride that a country is doing better at vaccination than its rivals, or pride in having produced the vaccine itself, a *Buy American* kind of effect). This may also have generated local political capital and global prestige. However, such rivalry may also have negative impacts if other countries are denied vaccine access or are only supplied with vaccines form a country or company that they do not trust.

### 2.6. Integrating the Full Behavioral Intervention Mix

In recent decades, behavioral science has identified a taxonomy of values and a toolbox of strategies for eliciting behavioral response [30,31]. These include a variety of persuasive communication and “nudge” or default option approaches to encouraging behavior adoption, overcoming barriers, and maintenance of behavior change. These include not only “nudges,” but “hugs,” “shoves,” and “smacks” all based on research and insight.

**Hug =** A hug is an active, positive exchange involving incentives to reward people who choose to do a particular behavior. An example would be access to a shopping mall if you have a vaccine certificate.**Smack =** A smack is an active, negative exchange that uses disincentive punishment for conscious/considered behaviors. An example would be not being allowed into a shopping mall without proof of vaccination.**Shove =** A Shove is a passive, negative exchange that involves a disincentive punishment on the automatic/unconscious decision. An example would be being the inconvenience required to take multiple antibody tests before being able to work if you are not vaccinated.**Nudge =** A nudge is a passive, positive exchange that involves incentives to reward people who do a particular behavior. For example, a default scheme for the whole population calling people to be vaccinated

See Figure 3 for a graphic representation of these four types of interventions.

Most effective strategies will employ a mix of these intervention types and forms together with communication plans, educational efforts, call and recall systems, community engagement strategies and vaccine service delivery systems that are efficient and user friendly.

### 2.7. Vaccine Service Marketing Strategy Considerations

Developing effective vaccine promotion branding in both the short and long-term requires co-design with citizen’s input. Co-design calls for formative research, and the importance of continuous improvement driven by customer/patient and staff feedback. It also calls for developing patient/customer champions and advocates to recruit others, build trust, and create a sense that there is social movement to promote vaccination [33,34], a sense of momentum toward re-normalizing society through large scale efforts that are greater than each of us individually [35].

In order to develop such a strategy, formative research to understand audience barriers and potential supports for demand creation is crucial. The aim of the formative research is to understand from each target audience:Knowledge about COVID-19 and the safety and efficacy of the vaccine.Barriers, beliefs and taboos relating to vaccine acceptance, including fears and rumours about COVID-19 vaccine.Social norms about vaccination among children and adolescents—how prevalent and acceptable is it perceived to be?Who are the opinion leaders about vaccination?Who are the influential people for each target group and opinion leaders?Who are potential partners and allies that can help to support the campaign, including health care providers?If there are gender dynamics within the household and community that would affect vaccine acceptance and getting vaccinated.What are the demographics of and social determinants of vaccination or of wider related issues like access to healthcare or school?Where and how they communicate about the issues that matter to them? With whom?Where and through what communication channels (e.g., social media)?Which channels and people make them most satisfied?What inputs they have for the design of a demand generation strategy and plan.

### 2.8. Vaccination Brand Development and Strategy

Under a single name and identity, or a set of core values and principles, a brand unites all of a product, service, organization, or behavior change promotion activities [36,37]. It represents key aspects of the consumers’ lives and connects the core components of the brand to its specific executions, such as promotion (advertising), placement, pricing, and product (4 Ps).

Vaccination can be branded, following behavioral branding principles used in social marketing. Brands in the commercial sector build relationships [38] between products, services, organizations and consumers. Similarly, ‘behavioral brands’ are the relationships that individuals form with behaviors and lifestyles that embody multiple behaviors, and can be measured by the associations they form with health behaviors and lifestyles [19]. In the health sector, the concept of behavioral branding has become a major component of social marketing campaigns to change a wide range of behaviors from HIV prevention to tobacco use prevention and physical activity promotion [39]. Brands in both the commercial and non-commercial sectors can also apply to organizations, and upstream factors that promote organizational impact and wellbeing. Health branding—building positive associations with healthy behaviors and lifestyle choices such as vaccination—is the primary strategy by which commercial marketing is applied in social marketing [19].

A brand’s strength is generating “demand” for behavior change, making it desirable for the specific population to want to change, such as to move from being hesitant toward vaccination to getting the COVID-19 jab. There is also a key supply-side component to behavior change. On the supply side institutions, policies, financing, and quality assurance must all be in place. COVID-19 vaccine promotion branding should also include actions focused on partnership building and solution generation to address supply-side logistical issues like ensuring that supply is available and easy to access.

### 2.9. Recommended Generic Vaccine Brand Building Strategy

Based on our previously published a set of key guidelines for COVID-19 vaccine promotion [14], we now argue that these broad steps can be effectively operationalized and implemented to overcome vaccine hesitancy following the specific strategies outlined above. To carry out these overall strategies, we outline here an approach to demand creation based on a branding strategy.

The program should be designed based on in-depth formative research following methods shown effective in producing demand for many other vaccines demand [20,40,41]. The aim of the formative research is to understand from each target audience:Knowledge about COVID-19 infection, prevention methods, and the safety and efficacy of specific COVID-19 vaccines.Barriers, beliefs and taboos relating to vaccine acceptance, including fears and rumours about the COVID-19 vaccine.Social norms about vaccination among population groups who are defined as hesitant or resistant—how prevalent and acceptable is it perceived to be?Who are the opinion leaders about vaccination?Who are the influential people for each target group and opinion leaders?Who are potential partners and allies that can help to support the campaign in government and civil society?If there are gender dynamics within the household and community that would affect vaccine acceptance and getting vaccinated.What are the demographics of hesitant individuals and what are the social determinants of vaccination or of wider related issues like access to healthcare?Where and how they communicate about the issues that matter to them? With whom? Where/through what channels? Which channels and people make them most satisfied?What inputs they have for the design of a demand generation strategy and plan.

To do this, rounds of focus groups, semi-structured interviews and population level attitude, awareness belief and reported behavior should be organized and conducted

After the completion of the formative research, a demand generation program plan should be developed. This should include a detailed strategic and operational plan showing all steps, behavioral strategies, message strategies and channels, target audiences, a theory of change model, a stakeholder management plan, monitoring and evaluation plan, methods and measurement tools, media communication plan. It will be informed by feedback and co-creation activities in the beginning, and feedback and approval before collateral finalized. Adjustments may need to be made to specific aspects of the program during the implementation and will account for this.

The demand generation plan should include five primary strategies:Raising public awareness of COVID-19 vaccination through social media, radio, TV, online, and print advertising (billboards and posters) and other channels, processes identified during the formative research that the team deems feasible.;Pre vaccination and point of decision promotion in health care facilities through informational didactic videos, posters, and pamphlets that explain the benefits, safety, and dispel myths about the COVID-19 vaccine (NOTE: This same information will be available on a campaign website created and hosted by the research team); Other forms of behavioral influence should also be used such as promoting vaccination as a social norm, using Hugs, Shoves, Smacks and Nudging strategies and taking vaccines delivery services out into the community to improve access.COVID-19 vaccination ambassadors and partners including influential persons, organizations, businesses in local communities who will be recruited and trained to deliver pro-vaccination information within their towns and villages; These groups will also be used to provide continuous feedback about how well the program is being delivered, its impact and how it can be improved.Educational entertainment (edutainment) in the form of a reality-based video and radio series that depicts a family and community with vaccination-age children making decisions to get vaccinated, to be delivered on TV, radio, and social media such as YouTube and Facebook and modelled on successful series developed on other topics such as HIV prevention and modern cookstove purchase [42,43].Service design strategy that ensures that access to vaccination is an easy low-cost experience that instils confidence in the service providers and vaccine. Citizens should be involved in helping to design and evaluate the success of systems to call people for vaccination, administer vaccines and follow-up.

This plan has two overall purposes: (1) public education to increase knowledge, beliefs in safety and efficacy, and reduce hesitancy to receive the HPV vaccine (primarily strategies 1–2); and (2) influencing vaccine uptake behavior by creating a positive social norm that vaccination is widespread and socially acceptable). The program should be implemented in partnership with a local media organization and be based on results of the formative research, which should be reviewed in depth in a collaborative process including the research team, media organization, government public health officials. The result of this process will be a consensus planning document on specific implementation tactics for each of the four strategies.

While the specifics of the demand generation plan must wait for the formative research input, the following proposed tactics are likely to be reflected in most intervention programs. First, raise public awareness of the coming availability and places to access available vaccines, and who needs to come forward through radio, TV, online, and outdoor advertising in the selected locations, and community engagement programs such as asking for volunteers to assist with the program roll out. Ads will be factual and will have a recognizable brand identify, logo, colour scheme, and distinctive stylistic features that will become widely recognized. Ads should feature representations of people who are being called for vaccination.

Second, multiple communication channels should be used to promote the benefits, safety, and efficacy of the vaccine, and be made widely available in health care facilities in the selected locations. A common and consistent brand identity, logo, should be developed and used on all vaccine related promotions to ensure it is recognized as coming form and trusted source.

Third, recruit influencers, such as artists, social media influencers, business and religious leaders, teachers, medical staff, parents, politicians, and local celebrities to be ambassadors for vaccination. Given ambassadors support to promote the objectives of the program and the task of assisting in raising awareness, and also serve as influencers to generate public support.

Fourth work with media firms and social media firms to develop informational and educational entertainment programs focused on the benefits and rewards associated with vaccine uptake. These should focus on creating an appealing persona for the vaccination uptake brand through characters who embody values appealing to the audience who decide to take the COVID-19 vaccine. These role models will encourage emulation by the audience and break down barriers and increase trust. All materials and interventions developed will be pre-tested including graphics, text, educational and video content with community members before deployment.

## 3. Discussion

Vaccine hesitancy is on a continuum, ranging from cautious to actively resistant. Audiences must be addressed based on where they stand on that continuum, and understanding complex segmentation factors including demographics, social norms, and market factors promoting and serving as barriers to vaccination must be well understood. Audience research and insight is a crucial factor in designing interventions to overcome hesitancy to vaccine acceptance, COVID-19 vaccination in particular given the attention paid to it since 2020 and the large amounts of conflicting and often mis- and dis-information that has been disseminated about the disease and its prevention and treatment [44].

Behavioral sciences, and social marketing both offer strategies that can be effectively employed to overcome vaccine hesitancy [45]. This includes the full tool kit of intervention strategies, based on the four Ps of marketing, and insights gleaned from psychology, economics, and persuasive communication [46]. It is crucial to recognize that vaccine acceptance and hesitancy occurs in a marketplace of ideas, which implies that competition analysis (i.e., understanding and designing interventions to address the appeal of anti-vaccination messages) is a central strategy.

Branding of vaccination behavior and specific COVID-19 vaccine brands will therefore be crucial to overcoming hesitancy. Branding is a strategy to create a clear-cut identity and a set of values not only for the behavior of being vaccinated, but also for specific vaccine brands. As the pandemic moves from a situation in which supply is low and only the enthusiastic and eligible (e.g., front line workers and the elderly) are eligible for vaccination, to one in which supply is greater and many hesitant groups become eligible, branding in both senses will become crucial.

To create effective brands, we need to bring them to life using a full range of intervention strategies including creating messages framed through personas (characteristics of trusted individuals for the hesitant audiences) and creating choice architectures (i.e., frameworks of choice in the real world in which choosing to be vaccinated for COVID-19, and trusting in a specific vaccine brand) that are favorable to increasing vaccination rates (i.e., making the right choice).

We have outlined a set of theory-based strategies to develop a marketing strategy and a COVID-19 vaccination brand for use both in the early stages and latter stages (where hesitancy will be a significant barrier) of the rollout. Specifically, we view both vaccination behavior and the specific vaccines as brands competing within a marketplace of ideas and behaviors. These brands must have characteristics that make them appealing and trusted to all consumers, but specifically to the hesitant and resistant. This requires attention to the choice scenarios that are present in high and low supply and demand circumstances. The Porter market forces model provides a framework for this approach.

Following that overall framework, we have described the competition strategy that has proven effective in other relevant behavior change situations and have argued for an approach which includes reward and incentive structures. The approach applied in psychology and behavioral economics of creating environmental incentive structures, sometimes referred to as “nudges,” may be used in a full active decision model using both active and passive tactics. We argue for a competition analysis and creation of a brand aimed at employing these tactics in a targeted fashion demanding on the choice scenario in a specific context.

Finally, we have described a specific approach to addressing hesitancy within a structured campaign format. This involves creating a vaccine uptake brand based on a comprehensive strategic plan informed by formative research with and insight into hesitancy among the population of interest. The plan includes behavioral strategies, message strategies and channels, target audiences, a theory of change model, a stakeholder management plan, monitoring and evaluation plan, methods and measurement tools, media communication plan. It employs continuous feedback and co-creation activities throughout the program. There are five key steps in such an approach: raising awareness of vaccine availability, safety, and benefits; point of decision prompts in health care facilities; use of influencers including COVID-19 vaccination ambassadors (to build trust); educational entertainment to normalize vaccination; and a service delivery strategy.

These combined approaches, representing the full intervention mix, have been effective in other vaccination contexts to reduce hesitancy. We recommend continuous monitoring and evaluation of these strategies to build the evidence base and identify specific components of the intervention that may be more or less effective in the context of COVID-19, which has specific dimensions in terms of severity, duration, and mis- and dis-information that is present in the media and social environments.

### Limitations

This review has not addressed in detail how existing vaccine products are being branded by the governments and companies that have brought them into existence and the consequences of this branding activity. We have not addressed the unfolding competitive environment that is emerging amongst these vaccines despite the efforts of the COVAX system or the impact of new entrants. We have also not covered issues such as vaccine nationalism and how this intersects with vaccine diplomacy. Vaccine price, transportability, storage, and the probable need for booster shots within a year or so are also factors that space has not permitted us to address in this paper despite their likely high impact of vaccine promotion and branding. These are all urgent issues requiring further research and review.

## 4. Conclusions

In conclusion, it is crucial for COVID-19 vaccination promotion efforts to be clear as to what stage they are in the process at all times, who is being targeted, and to have insight into the basis of their willingness or hesitancy to be vaccinated. The context and choice architecture in which the demand promotion program is operating (high choice/low choice) sets the stage for which components of the intervention mix should be used, when, and using what specific implementation tactics. In general, there is a need for a comprehensive approach, and for a strategic operational planning process that looks at both short and long-term issues given that this is a long-term fix. We present a specific set of strategies for implementation of a COVID-19 vaccine promotion strategy based on these evidence-based approaches.

## Figures and Tables

**Figure 1 vaccines-09-00319-f001:**
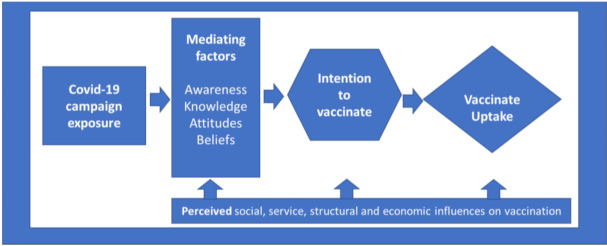
Social, service, structural, and economic influences on vaccination.

**Figure 2 vaccines-09-00319-f002:**
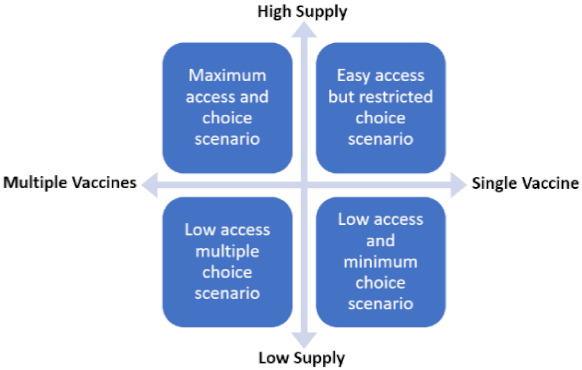
Vaccine Choice Scenarios.

**Figure 3 vaccines-09-00319-f003:**
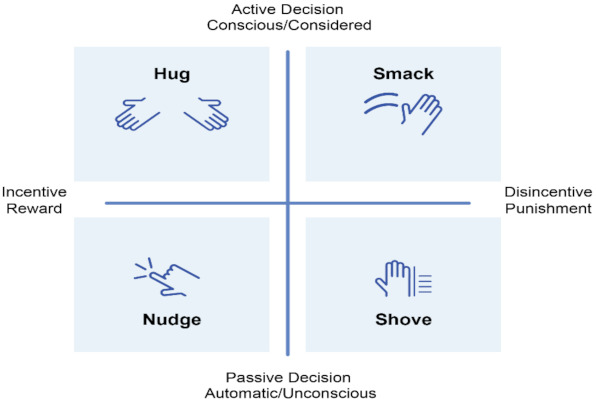
Reward and Incentive Scheme [32]. Copyright permission.

## Data Availability

Not applicable.

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
