# Peer review of "Demand Creation for COVID-19 Vaccination: Overcoming Vaccine Hesitancy through Social Marketing"

_vaccines, 2021, doi:10.3390/vaccines9040319_

Round 1

Reviewer 1 Report

This is a highly relevant and pertinent discussion piece about vaccine hesitancy which is a major problem at the moment.  Other than minor editorial corrections, the main suggestion I would make, in order to make the paper right up to date, is to add a paragraph about the latest situation in Europe with vaccine hesitancy. This should be referenced from media, but in summary a few weeks ago France stopped using the AZ vaccine because there was no evidence from clinical trials (because they weren't included) of efficacy against over 65s which was misinterpreted as the vaccine is not effective against over 65s, which led to them stopping use. Then more recently concern over possible side effects led to cessation of use until the European medicines regulator declared it safe, but the damage to the vaccine programme in Europe is significant in terms of public confidence.

Author Response

Thank you. We have made minor editorial corrections and have added a paragraph about the situation in Europe noted by the reviewer.

Reviewer 2 Report

I was invited to revise the paper entitled "Demand creation for COVID-19 vaccination: Overcoming vaccine hesitancy through social marketing". This review aimed to summarize possible strategies against covid19 vaccine hesitancy, highlightening factors that can influence the vaccine uptake.

The topic is very interesting and this paper is relevant, improving the knowledge in this field. 

Introduction is well presented and clearly describe the study background. In particular, the description of the importance of public figures in vaccine promotion is relevant and well focused.

Methods are adequate and well described the model proposed by Authors. In addition, figures are clear and facilitate the reading.

Discussion deeply describe the study results and are easy to read.

I only suggest to discuss about vaccine hesitancy among healthcare workers. It can be a factor that strongly influence the vaccine uptake both among HCW and public patients. 

Author Response

Thank you. We added a brief discussion of the need to promote vaccine uptake among HCW as part of a strategic approach to overcoming hesitancy. HCW are critical to building trust in the vaccine and serve as role models for patients.